# Characterizing the Spatial Uniformity of Light Intensity and Spectrum for Indoor Crop Production

László Balázs [1], Zoltán Dombi [2], László Csambalik [3] and László Sipos [4,5,*]

1 Department of Microelectronics and Technology, Kálmán Kandó Faculty of Electrical Engineering, Óbuda University, 17 Tavaszmező út, 1084 Budapest, Hungary; balazs.laszlo@kvk.uni-obuda.hu
2 Department of Mechatronics, Optics and Mechanical Engineering Informatics, Faculty of Mechanical Engineering, Budapest University of Technology and Economics, 3. Műegyetem rkp., 1111 Budapest, Hungary; domzolab@gmail.com
3 Department of Agroecology and Organic Farming, Institute of Rural Development and Sustainable Production, Hungarian University of Agriculture and Life Sciences, 39-43 Villányi út, 1118 Budapest, Hungary; csambalik.laszlo.orban@uni-mate.hu
4 Department of Postharvest Science, Supply Chain, Commercial and Sensory Evaluation, Institute of Food Science and Technology, Hungarian University of Agriculture and Life Sciences, 39-43 Villányi út, 1118 Budapest, Hungary
5 Centre for Economic and Regional Studies, Loránd Eötvös Research Network, Institute of Economics, 4. Tóth Kálmán utca, 1097 Budapest, Hungary
* Correspondence: sipos.laszlo@uni-mate.hu

**Abstract:** Maintaining uniform photon irradiance distribution above the plant canopy is a fundamental goal in controlled environment agriculture (CEA). Spatial variation in photon irradiance below the light saturation point will drive differences in individual plant development, decreasing the economic value of the crop. Plant growth is also affected by the spectral composition of light. So far, little attention has been paid to the quantification of the spatial variability in horticultural lighting applications. This work provides a methodology to benchmark and compare lighting installations used in indoor cultivation facilities. We measured the photon irradiance distributions underneath two typical grow light installations using a $10 \times 10$ measurement grid with 100 mm spacing. We calculated photon irradiance values for each grid point for 100 nm-wide blue, green, red and far-red wavebands covering the 400–800 nm range. We showed that the generally used uniformity metric defined as the minimum to average ratio of PPFD is not appropriate for the characterization of light uniformity in horticultural lighting applications. Instead, we propose to normalize photon irradiance to the maximum, analyze the histograms constructed from relative photon irradiance values and consider the light response of the cultivated crop while comparing the performance of CEA grow systems.

**Keywords:** sole-source lighting; spectroradiometer; lighting characteristics; crop growth model; vertical farm productivity

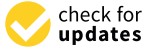



## 1. Introduction

Growing plants in a controlled environment, such as in vertical farms or plant factories, has many advantages over traditional horticulture [1]. Indoor cultivation provides optimal conditions for plant development allowing year-round cultivation at any location around the globe. Factors affecting plant growth are tightly controlled; therefore, crop yield will be predictable and independent of local weather fluctuations [2]. Vertical farms can be deployed close to end-users, reducing the environmental impact of transportation. Application of soilless cultivation methods [3] reduces or eliminates the need for pesticides and herbicides, providing healthy fresh products for the consumers [4].

High energy demand hinders the spread of sole-source lighting applications. The electricity consumption of electric lighting is a key factor influencing the economics of indoor

farming [5]. Although LEDs have significantly increased lighting efficiency relative to gas discharge technologies in recent years, the efficacy of current LED packages has already reached 80% of the theoretical maximum; therefore, there is limited improvement opportunity left in the light source efficacy [6]. Because of the high proportion of lighting-related electricity consumption in the total energy demand [1], optimizing lighting conditions for maximum crop yield and minimum energy consumption is a key challenge for the designers of CEA applications.

Spatial and temporal variations in light can be advantageous or limiting; shade plasticity is different on a plant species level [7]. Many traits related to plant morphology, anatomy and physiology including growth and reproduction increase with light intensity and go to saturation beyond a certain threshold [8]. In a natural environment, the light intensity may exceed the light saturation point above which excess light is no longer utilized by the crop. Too much sunlight may trigger degradation mechanisms in plants [9]; therefore, some crops need to be protected against harsh radiation.

Besides horizontal PPFD distribution, spatial and temporal changes in within-canopy vertical light environments also play a key role in case of tall crops or in a forest canopy [10–12]. Yield and quality of crop depend both on the spectral composition and the intensity of photon irradiance, often referred to as the qualitative and quantitative parameters of light [13,14].

For photosynthesis, plants utilize light mainly in the 400–700 nm spectral range called Photosynthetically Active Radiation and abbreviated as PAR. Among other factors, the rate of photosynthesis is determined by the photon irradiance within the PAR wavelength range, a quantity generally known as Photosynthetic Photon Flux Density (PPFD) expressed in $\mu mol \cdot s^{-1} \cdot m^{-2}$. Following the recommendations of the International Commission on Illumination [15], in this paper, we use the term "photon irradiance" as the synonym of photon flux density concerning the quantity measuring the number of photons incident on a surface per unit time and unit area, but we will keep PPFD notation for the photon irradiance within the PAR wavelength range as its use is common in the horticultural literature.

In real horticultural systems, light is not uniformly distributed across the illuminated surface; therefore, plants are not exposed to the same lighting conditions. Spatial variation in photon irradiance will drive differences in individual plant development decreasing the yield and commercial value of the crop. Simple practices, such as random or systematic relocation of plants [16,17] or rotation of trays [18,19] within an experimental design might reduce experimental error arising from light inhomogeneity; obviously, this is not a viable option for commercial CEA production.

Traditionally, evaluations of the lighting environment have focused on the quantitative aspects of light. Both in scientific experiments and tests in plant production facilities, PPFD has usually been provided as a single measurement value or as the average of a few measurement points. Relatively little attention has been paid to the quantification of PPFD uniformity over the plant growing area.

In the case of gas discharge lighting, it was a valid assumption that the type of light source (high-pressure sodium, metal halide, fluorescent) already defined the spectral distribution of the irradiance at any location of the illuminated area. Spatial deviations in photon irradiance were attributed merely to the variation in the light intensity, and the relative spectral irradiance (the ratio of the peaks in the spectra) was regarded as constant.

LED technology brought new opportunities and also new complexity in horticultural lighting [20]. The LED luminaires incorporate discrete light emitting diodes radiating at different wavelengths. In the majority of LED grow lights, there are at least two different types of narrow band (NB) LEDs emitting in the blue (typically at ~450 nm) and in the deep red (typically at ~660 nm) wavelengths, respectively, matching the absorption maxima of chlorophyl molecules.

Combination of narrow band blue and red illumination does not necessarily provide the optimum condition for indoor crop production [16–22]. One design approach to expand the wavelength range of the spectrum is to include phosphor-converted white LEDs in the

luminaire. White LEDs have a narrow emission peak in the blue range and a broad emission peak filling the green gap between the blue and red wavebands in the visible spectrum.

Another approach to enrich the emission spectrum is to increase the variety of narrow band emitters in the lighting fixture. Superposition of a big number of NB LEDs with emission peaks ranging from the violet up to the far-red enables the resulting spectrum to be tailored to the specific need of the crop.

Additive color mixing of the light emitted by the individually separated LED chips occurs only partially, if at all, in the luminaire. In the case of small separation between the luminaire and the illuminated surface, there is an appreciable spatial variation in the spectral irradiance distribution on the illuminated plane depending on the actual position of discrete LEDs and the location in which the spectral irradiance distribution was measured.

Plant growth is influenced not only by the actual value of PPFD (and through the photoperiod by the daily light integral, DLI) but also by the spectral quality of photon irradiance [16–22]. Spatial variation of the light spectrum has not been the focus of recent studies despite the evidence that spectral inhomogeneity, especially changes in red/blue [17,20] and red/far-red [18,21,22] photon irradiance ratios, does have a significant influence on the rate of photosynthesis.

For the proper characterization of horticultural lighting applications, we need to know how the spectral distribution of photon irradiance varies over the illuminated surface. Direct comparison of spectral distributions consisting of several hundred measurement points would be difficult to handle. For most practical applications, it is reasonable to measure photon irradiance in 100 nm-wide bins within the 400–800 nm wavelength range. This is in line with the latest technical requirement for LED-based horticultural lighting products [23] to break down photon irradiance distribution interpolated to integer wavelength values with 1 nm resolution for blue, green, red and far-red wavebands corresponding to 400–499 nm, 500–599 nm, 600–699 nm and 700–800 nm, respectively.

Although lighting conditions are considered to be one of the key control parameters in CEA applications [24–35], there is still no standardized methodology to characterize the spatial variability of photon irradiance distributions. Quantification of photon irradiance distributions is important both from scientific and economic perspectives. As a common practice, PPFD is measured by quantum sensors which does not provide spectral information on the illumination [34]. Spatial distribution of light intensity as an uncontrolled noise factor limits the reproducibility of experiments and reduces the statistical significance of measurement results. In cultivation facilities, light inhomogeneity can reduce crop yield and production efficiency. The lack of a universally accepted protocol is considered as a major issue while comparing results of plant photobiological research studies [31,34]. Pan et al. have published methods to map intensity and spectral characteristics in 3D space which can provide a full characterization of the lighting environment relevant to plant growth [36], but the near-field goniometry proposed has been considered to be complicated and expensive for general use in horticultural lighting [34].

In this paper, we present a methodology for the characterization of photon irradiance distributions in plant cultivation facilities. The measurement procedure is relatively simple and can be carried out by affordable spectroradiometers. To illustrate the key points of data collection and processing, we compare two illumination scenarios widely used in controlled environment agriculture. The first is a canopy lighting with a single, high output luminaire placed at a relatively large distance from the work plane of the application. The second one is a lighting setup typically used in vertical farms where several lightbars of small footprint are placed close to the illuminated surface. Both lighting environments have similar average PPFD values and red/blue peak emission ratios; therefore, following the current practice, one would conclude that the two scenarios provide close to identical conditions for plant growth. By analyzing the spatial variability of irradiance spectra, we highlight important differences between the two lighting environments. Our objective is to show the limitations of current practices and present a new approach, tailored to the needs of agriculture, to fully characterize photon irradiance distributions.

## 2. Materials and Methods

We made two experimental arrangements which will be noted as illumination settings (A),(B) throughout this paper. In illumination setting (A), we used a 630 W KindLED K5 XL1000 LED grow light originally designed to replace 1000 W high pressure sodium lamps. The hanging height of the luminaire above the work plane of the measurement was 1612 mm. The 668 mm $\times$ 495 mm housing incorporated 320 light-emitting diodes arranged in 16 rows and 20 columns. The luminaire contained a mix of 3 W and 5 W narrow band LEDs with emission peaks at 410, 435, 445, 465, 495, 595, 605, 630, 660, 675 and 735 nm as well as one type of broadband white phosphor LED. Each LED was equipped with a secondary lens narrowing the beam angle of the emitted light.

In the second configuration, illumination setting (B), 8 parallel lightbars developed for vertical farms (TUAS VFP Tungsram Agritech) were positioned in the horizontal plane 200 mm above the illuminated surface. The total power of the 8-lightbar system was 402 W. The length of the light emission window of the lightbar was 1102 mm. The 4-channel LED luminaire incorporated 3 types of narrow band LEDs with emission peaks at 445, 660 and 735 nm and one type of phosphor-converted white LED. No secondary optics were applied in the lightbars.

The luminaires were operated from a stable output AC power supply (California Instruments 1501IX-LKM). The first measurement was taken after a 30 min warmup period to ensure constant photon flux emission throughout the measurement. Following a pretest measured only in 5 points of the grid, the photon radiance of the luminaire was adjusted to establish 200 µmol m$^{-2} \cdot$s$^{-1}$ average photosynthetic photon flux density on the work plane in both arrangements. We did not make any further attempt to fine-tune the photon irradiance distribution across the illuminated surface.

Irradiance spectra were recorded at 100 different points of the illuminated plane using an AvaSpec$-$2048 spectroradiometer. The optical fiber connected to the spectrometer was equipped with a cosine corrector to collect light from the 180° field of view. To position the light probe of the spectrometer measurement grid made of a 5 mm thick black PVC foam board was used. The lateral dimensions of the sheet were 1100 $\times$ 1100 mm. The sheet had 2 $\times$ 2 slots close to two opposite sides for positioning and fixing to a table in the horizontal plane. The sheet was perforated to form a 10 $\times$ 10 grid with 100 mm spacing between two adjacent holes. The cosine corrector tightly fit into the holes and determined both the lateral position and the height of the probe above the sheet.

The outline of the luminaires projected to the measurement grid is shown in Figure 1. For illumination setting (A), the axis of the grow light was aligned with the center point of the measurement grid. In illuminance setting (B), the leftmost and rightmost light bars were positioned above the first and last column of the measurement grid, whereas the remaining 6 lightbars were uniformly distributed between the two opposite sides of the grid, as shown in Figure 1B.

In each hole of the measurement grid, one irradiance spectrum was recorded between the 380 nm–780 nm wavelength range. From the discrete spectral irradiance values, we calculated the related photon irradiance ($E_p$) for five wavebands, noted as B, G, R, FR and PAR corresponding to 400–499 nm, 500–599 nm, 600–699 nm, 700–800 nm and 400–700 nm wavebands, respectively.

$$E_p = \frac{10^6}{hcN_A} \sum_{\lambda=\lambda_{min}}^{\lambda_{max}} \lambda E_e(\lambda) \Delta\lambda \tag{1}$$

In Equation (1), $E_p$ is the photon irradiance expressed in µmol$\cdot$m$^{-2}\cdot$s$^{-1}$, $h$ is the Planck constant, $c$ is the speed of light, $E_e(\lambda)$ is the spectral irradiance measured in W$\cdot$m$^{-2}\cdot$nm$^{-1}$ at wavelength $\lambda$ and $N_A$ denotes the Avogadro constant. $\lambda_{min}$ is the lower, $\lambda_{max}$ is the upper bound of the wavelength interval, and $\Delta\lambda$ is the wavelength difference between adjacent data points at $\lambda$. For each ($x,y$) point on the measurement grid, we determined the photon irradiance values related to the B, G, R, FR and PAR wavebands denoted by $E_B(x,y)$, $E_G(x,y)$, $E_R(x,y)$, $E_{FR}(x,y)$ and $E_{PAR}(x,y)$, respectively. In one illumination setting, 100 spectra

were measured, and considering the five wavebands, altogether $5 \times 100 = 500$ data points were generated.

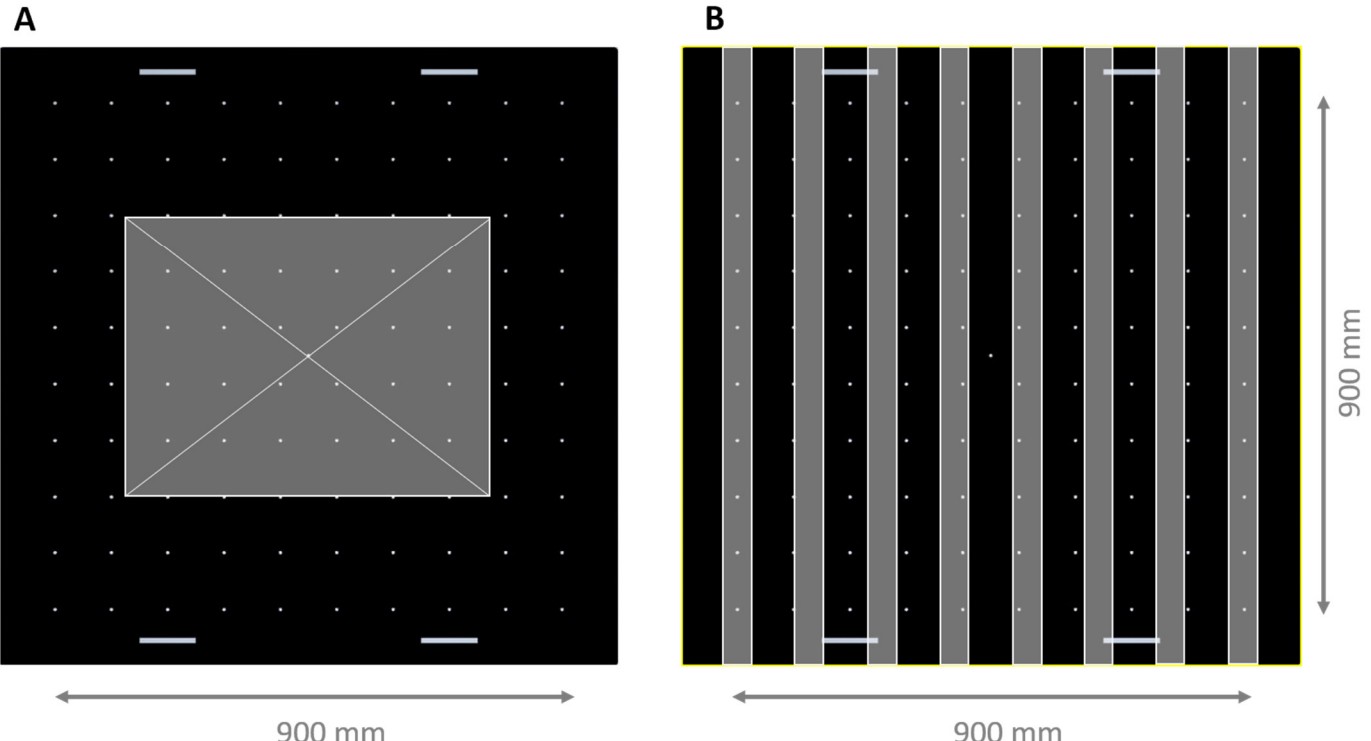

**Figure 1.** Schematic of the $10 \times 10$ measurement grid (black) with the outline of the luminaires (gray) for illumination settings (A),(B). The hanging heights of the luminaires were 1612 mm (**A**) and 200 mm (**B**) above the measurement plane. The distance between the grid points was 100 mm in both directions of the horizontal plane.

## 3. Results

Representative spectral irradiance distributions ($E_e$ ($\lambda$) measured in $\mu$mol m$^{-2}$ s$^{-1}$) for illumination settings (A),(B) are shown in Figure 2. Statistical parameters characterizing the spatial variability of the photon irradiance distributions are listed in Tables 1 and 2.

**Table 1.** Statistical parameters characterizing the photon irradiance distributions in the B, G, R, FR and PAR wavebands for illuminance settings (A),(B). The photon irradiance parameters ($E_{max}$, $E_{ave}$, $E_{min}$) have unit $\mu$mol·m$^{-2}$·s$^{-1}$.

| Setting | (A) | | | | | (B) | | | | |
|---|---|---|---|---|---|---|---|---|---|---|
| **Wave Band** | **PAR** | **B** | **G** | **R** | **FR** | **PAR** | **B** | **G** | **R** | **FR** |
| $E_{max}$ | 233 | 28.9 | 12.6 | 192 | 6.62 | 250 | 35.7 | 39.3 | 176 | 42.0 |
| $E_{ave}$ | 214 | 26.2 | 11.4 | 176 | 6.01 | 198 | 28.3 | 30.6 | 139 | 31.7 |
| $E_{min}$ | 186 | 22.8 | 9.7 | 153 | 5.21 | 119 | 17.5 | 18.4 | 83.2 | 18.4 |
| $U_o$ | 0.87 | 0.87 | 0.85 | 0.87 | 0.87 | 0.60 | 0.62 | 0.60 | 0.60 | 0.58 |
| $U_d$ | 0.80 | 0.79 | 0.77 | 0.80 | 0.79 | 0.48 | 0.49 | 0.47 | 0.47 | 0.44 |
| $P_{10}$ | 0.85 | 0.84 | 0.82 | 0.85 | 0.84 | 0.61 | 0.61 | 0.60 | 0.61 | 0.55 |
| 1-CV | 0.95 | 0.94 | 0.94 | 0.95 | 0.94 | 0.82 | 0.83 | 0.82 | 0.82 | 0.81 |

**Table 2.** Statistics calculated for photon irradiance ratios R/B, R/FR and B/G in case of illumination settings (A),(B).

| Setting | (A) | | | (B) | | |
|---|---|---|---|---|---|---|
| Ratio | R/B | R/FR | B/G | R/B | R/FR | B/G |
| $R_{max}$ | 6.84 | 29.9 | 2.4 | 5.45 | 4.86 | 1.0 |
| $R_{ave}$ | 6.71 | 29.3 | 2.3 | 4.92 | 4.40 | 0.9 |
| $R_{min}$ | 6.53 | 28.8 | 2.3 | 4.55 | 4.00 | 0.8 |
| $U_o$ | 0.97 | 0.98 | 0.99 | 0.92 | 0.91 | 0.91 |
| $U_d$ | 0.95 | 0.96 | 0.93 | 0.83 | 0.82 | 0.84 |
| $P_{10}$ | 0.97 | 0.97 | 0.94 | 0.86 | 0.85 | 0.88 |
| 1-CV | 0.99 | 0.99 | 0.99 | 0.96 | 0.95 | 0.96 |

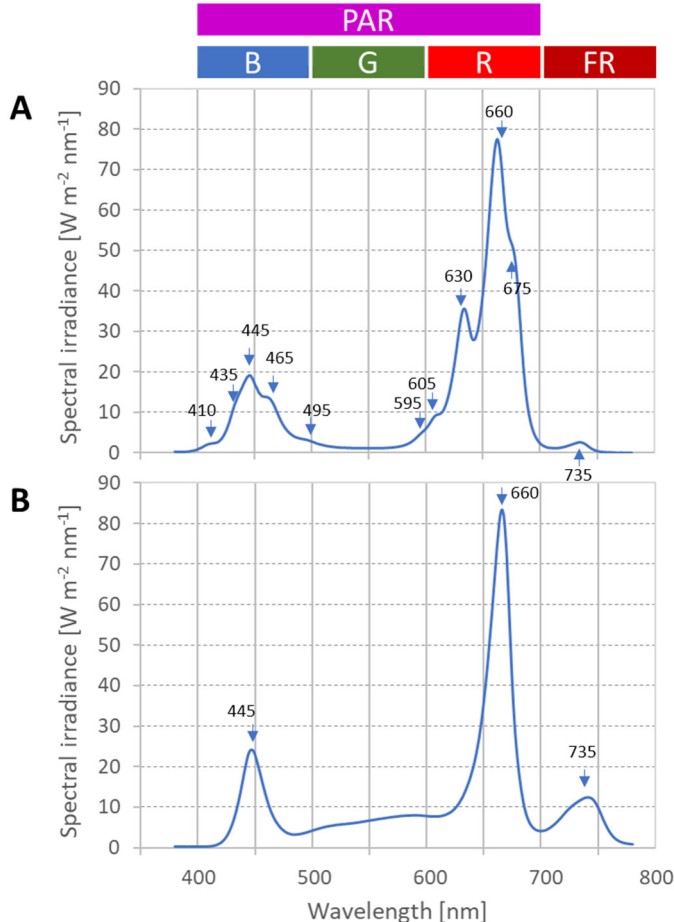

**Figure 2.** Representative spectral irradiance distributions measured for illuminance settings (A),(B). The numbers adjacent to the arrows indicate the peak wavelength of individual NB LEDs in nanometers. The spectrum is the superposition of 11 types of NB LEDs and one type of white LED in (**A**), whereas 3 types of NB LEDs and one type of white LED in (**B**).Note that the white LEDs have emission peak at 445 nm overlapping with the 445 nm NB LED spectrum. The broad emission between 500 and 600 nm is due to the phosphor emission of white LEDs. In (**B**), the large proportion of white LEDs explain the relatively high spectral intensity between 500–600 nm. The ranges denoted as B, G, R and FR represent wavebands for 400–499 nm, 500–599 nm, 600–699 nm and 700–800 nm, respectively. PAR stands for photosynthetically active radiation covering the wavelength range between 400–700 nm.

There seems to be little difference between the two environments if we restrict our analysis to the common practice of reporting one quantitative figure (PPFD) and qualitative (spectral) information as key light characteristics. The average PPFD values for settings (A),(B) are 214 µmol m$^{-2}$ s$^{-1}$ and 198 µmol m$^{-2}$ s$^{-1}$, respectively. There are two dominant

peaks in both spectral irradiance distributions at 445 nm and 660 nm as shown in Figure 2. The ratios of the red and blue peak heights are also close to each other: 4.0 in Figure 2A and 3.5 in Figure 2B. These data do not provide information on the spatial variability of photon irradiance which can be the main source of difference between the performance of the cultivation systems. Therefore, in our analysis we go beyond the common practice and provide more details by analyzing irradiance spectra and evaluating photon irradiance data integrated in the 100 nm-wide wavebands to account for the spatial variability of the illumination.

### 3.1. Analysis of Spectral Irradiance Distributions

In Figure 2, arrows indicate the peak wavelengths of the discrete NB LED types incorporated into the luminaires. In setting (A), there were 11 different types of NB LEDs and 1 type of white LED in the lighting fixture. The emission spectra of the individual NB LEDs have ~20 nm spread about the peak wavelength at full width of half maximum. The relative height of an individual NB LED emission reflects the proportion of the radiated power of the specific LED type relative to the total radiated power of the luminaire. The adjacent NB LED peaks overlap, resulting in a wide emission band exhibiting shoulders in the 400–500 nm and the 580–700 nm range.

The phosphor-converted white LED has a narrow emission at 445 nm and a broad emission with a maximum at 567 nm. The 445 nm irradiance is a superposition of the irradiances coming from the 445 nm NB LED and the 445 nm emission of the phosphor-converted white LED in Figure 2B. On top of the superposition of white and NB 445 nm irradiance, the contributions from the NB LEDs with adjacent peak wavelengths should also be considered in Figure 2A.

The broad emission in the green waveband between 500–600 nm is exclusively due to the phosphor emission of white LEDs in setting (B). In luminaire (A), the tails of the 495 nm, 595 nm and 605 nm NB LED peaks have contribution to the G region along with some white phosphor emission. The high intensity in the 500–600 nm wavebands of luminaire (B) relative to luminaire (A) is due to the high proportion of white phosphor LEDs in the lightbars of setting (B).

In Figure 2A,B, the highest peak was located at 660 nm as a result of the high proportion of 660 nm NB LEDs in both lighting equipment. Far-red emission at about 735 nm is apparent in both spectra, with higher intensity in the case of lighting (B) relative to luminaire (A). Far-red is outside the PAR range consequently photon irradiance beyond 700 nm is excluded from the PPFD calculation. Far-red radiation affects the growth and development of many crops [21]; therefore, a recent publication proposes the extension of the PAR range by 50 nm to the 400–750 nm waveband [22].

Another approach towards a more detailed characterization of the lighting environment is to measure photon irradiance values in 100 nm-wide wavebands between 400 nm and 800 nm as described above. In this way, we use four independent quantitative parameters instead of a single PPFD figure to characterize the lighting conditions. The ratios of the photon irradiance values related to the B, G, R and FR range carry information on the spectral distribution. Using four wavebands instead of the single PAR range increases the granularity of data and can be regarded as a reasonable trade-off between the simplicity and accuracy of processing spectral variations.

### 3.2. Statistical Analysis of Photon Irradiance Data

Statistical parameters determined for all the five wavebands and two illumination settings (A),(B) are shown in Table 1. We start our analysis by comparing photon irradiance distributions in the PAR range (PPFD values).

Comparing the absolute values of $E_{min}$, $E_{ave}$ and $E_{max}$ listed in Table 1 does not provide sufficient information to assess the performance of the lighting systems. The average photon irradiance can be regarded as a setting parameter from the application perspective. The average value of the photon irradiance or PPFD depends on the type of the luminaire

(photon flux and optics) and the mounting height of the luminaire above the canopy. State-of-the-art luminaires have built-in dimming capability; therefore, the magnitude of the photon radiance emitted by the luminaire and hence the photon irradiance on the work plane can be fine-tuned in an application.

The uniformity parameter, $U_o$, defined as the quotient of the minimum and average photon irradiance, was 0.87 for illumination setting (A) whereas the overall uniformity was significantly lower, 0.60 in setting (B) indicating that photon irradiance distribution is more spread out below the average in the second case. $U_o$ behaves as a figure of merit, and we can conclude from the measured values that setting (A) is better than setting (B), but the numbers cannot indicate the scale of difference in performance because the minimum/average ratio does not carry information on the behavior of the distribution above the average. Therefore, we incorporated in Table 1 the diversity, defined as the quotient of minimum and maximum photon irradiance, $U_d = E_{min}/E_{max}$ as a more relevant measure for horticultural applications. The diversity is also a uniformity parameter of the photon irradiance distribution reflecting the relative range of the measured data set. The third uniformity parameter we calculated was the complement of the coefficient of variation (CV) defined as $1 - CV = 1 - \sigma/E_{ave}$. $\sigma$ denotes the standard deviation of the measured photon irradiance values. This parameter has been proposed to be used in horticultural lighting applications because it is not sensitive to the determination of the extreme values [24–26].

Visualization of the PPFD distributions is much more informative than comparing sheer numbers. If we divide $E_{PAR}(x,y)$ values by the maximum $E_{max,PAR} = \max(E_{PAR}(x,y))$, we obtain the relative PPFD distribution. The normalization enables the direct comparison of the two photon irradiance distributions as shown in Figure 3. The same color scale applies for both plots highlighting differences between the two illumination settings.

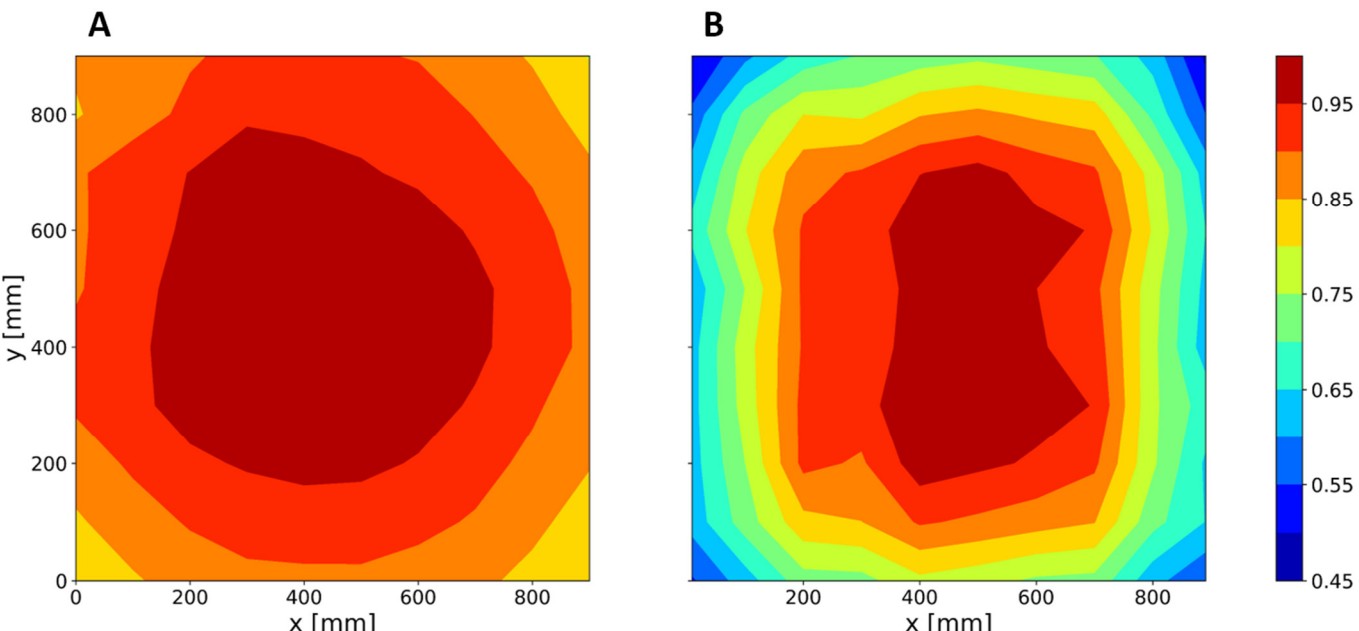

**Figure 3.** Relative photosynthetic photon irradiance (PPFD) distributions ($EPAR(x,y)/Emax,PAR$) measured for illuminance settings (A),(B). The mounting height of the luminaire was 1612 mm in (**A**), therefore only the flat middle portion of the emitted light determined the spatial distribution. Significantly lower PPFD values were measured at the boundaries of the target area in (**B**). The difference can be attributed to the low (200 mm) mounting height in setting (B) compared to setting (A).

We want to stress that differences are coming from the application design and do not qualify the luminaires as lighting products. The relative photon irradiance distribution is ultimately influenced by three factors: the horizontal arrangement of the LEDs over the

illuminated plane, the optical elements of light shaping and color mixing, and the mounting height of luminaires above the illuminated plane.

The edge effect is more pronounced in the case of vertical farm setting (B), where multiple luminaires were close to the work plane. In the canopy lighting scenario setting (A), the large area luminaire is at a relatively high distance from the work plane; therefore, only the flat middle portion of the emitted light beam contributes to the lighting task and the rest of the photons are lost from the application perspective. Both settings have their pros and cons: illumination setting (A) provides better uniformity, whereas illumination setting (B) ensures that a higher portion of photons is received by the target area.

Although the quotient of the photon flux incident on the work plane and the total photon flux emitted by the luminaire is an important efficiency measure of the lighting application, we restrict our discussion only to the system efficiency differences arising from the differences in photon irradiance distribution.

To quantify the information incorporated in the contour plots, we counted the number of data points falling within the bins of color scale and created histograms of PPFD distributions. The width of each relative photon irradiance bin was 0.05. The height of the bars in Figure 4 Indicates the proportion of data points within a bin relative to the total number of measurement values. A total of 29% of the data points were in the largest bin corresponding to the 0.95–1.00 bin in the case of illumination setting (A), and only half of this value, 15%, were found in the highest category in setting (B).

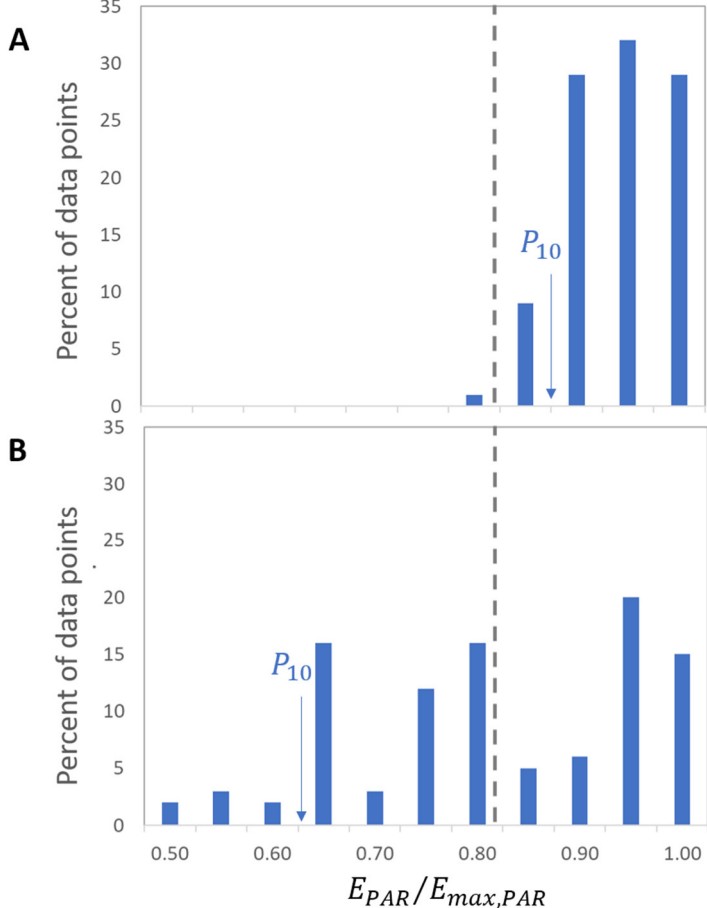

**Figure 4.** Histogram of relative photosynthetic photon irradiance (PPFD) values for illumination settings (A),(B). The labels on the horizontal axis are the upper bounds of the corresponding bins. The arrows indicate the 10th percentile (*P10*) of the data set. The dotted line represents a hypothetical specification limit. The width of the PPFD distribution is narrower in (**A**) compared to (**B**) in line with the contour plots in Figure 3.

The lowest value in the histograms corresponds to the diversity, $U_{d,PAR}$, which is equal to the relative range of the data set. Since the determination of the minimum is prone to measurement error, it is useful to determine an additional measure of variability [37], less sensitive to the data sampling errors, such as the $k^{th}$ percentile of the data set denoted $P_k$. In Figure 4, we indicated the 10th percentile, $P_{10}$, by arrow dividing the data set into two parts: 10% of data points have a value smaller than $P_{10}$ and 90% are greater than $P_{10}$. Table 1 shows the $P_{10}$ values calculated for each distribution.

### 3.3. Quantifying Spectral Variations

To account for the spectral variations over the illuminated surface, we calculated statistical parameters for the 100 nm-wide B, G, R and FR wavebands. Note that the sum of B, G and R location values are equal to the related PAR value. $U_o$ and $U_d$ exhibit small variations from the values determined for the PAR range, indicating slight deviations in the spectral distribution. The spectral output of the luminaires was constant throughout the experiment; therefore, local deviations in spectral distribution are due to improper color mixing of emitted light.

A better metric to assess the spectral uniformity of the photon irradiance distribution is the ratio of photon irradiances in two wavebands. Using the example of the red/blue ratio, we define $R_{R/B}(x,y)$, the quotient of photon irradiance in waveband R and waveband B, by

$$R_{R/B}(x,y) = \frac{E_R(x,y)}{E_B(x,y)} \tag{2}$$

Equation (2) defines a new distribution over the illuminated surface. We characterize the ratios with the same statistical parameters already presented for the absolute value of photon irradiance distributions: maximum ($R_{max}$), average ($R_{ave}$) and minimum ($R_{min}$), as well as the overall uniformity, diversity and $P_{10}$ of the photon irradiance ratios are listed in Table 2.

The location parameters of the distributions, $R_{ave}$, are determined by the color channel settings of the luminaires. The differences in the absolute values measured for illumination settings (A),(B) can be eliminated by adjusting the dimming level of the blue and the red channels of the luminaires. From an application design perspective, the difference between the diversities, $U_{d,R/B}$, $U_{d,R/FR}$ or $U_{d,B/G}$ is more relevant. In illumination setting (A), the difference between the lowest and highest value is in the range of measurement error. For illuminance setting (B), the ratios have greater spread and the 16–18% difference between the maximum and minimum values could have a substantial effect on a crop growth model.

To directly compare the distributions, we eliminated the differences resulting from different mean values by normalizing the photon irradiance ratios to their maxima. In Figure 5, we plotted the red/blue photon irradiance ratios $R_{R/B}(x,y)/\max(R_{R/B}(x,y))$ for illumination settings (A),(B). The contour plots share the same color scale to facilitate a direct comparison of the distributions. The R/B ratios reflect how the individual light-emitting diodes are distributed above the illuminated surface. The contour plot is also sensitive to the alignment of the luminaires providing valuable information for application engineers for the positioning and adjusting of the lighting system.

For the quantitative analysis of photon irradiance ratios, we determined the relevant histogram of R/B ratios. Figure 6 highlights that color mixing was more effective in illumination setting (A), where the distance between the work plane and the luminaire was greater than in the case of setting (B).

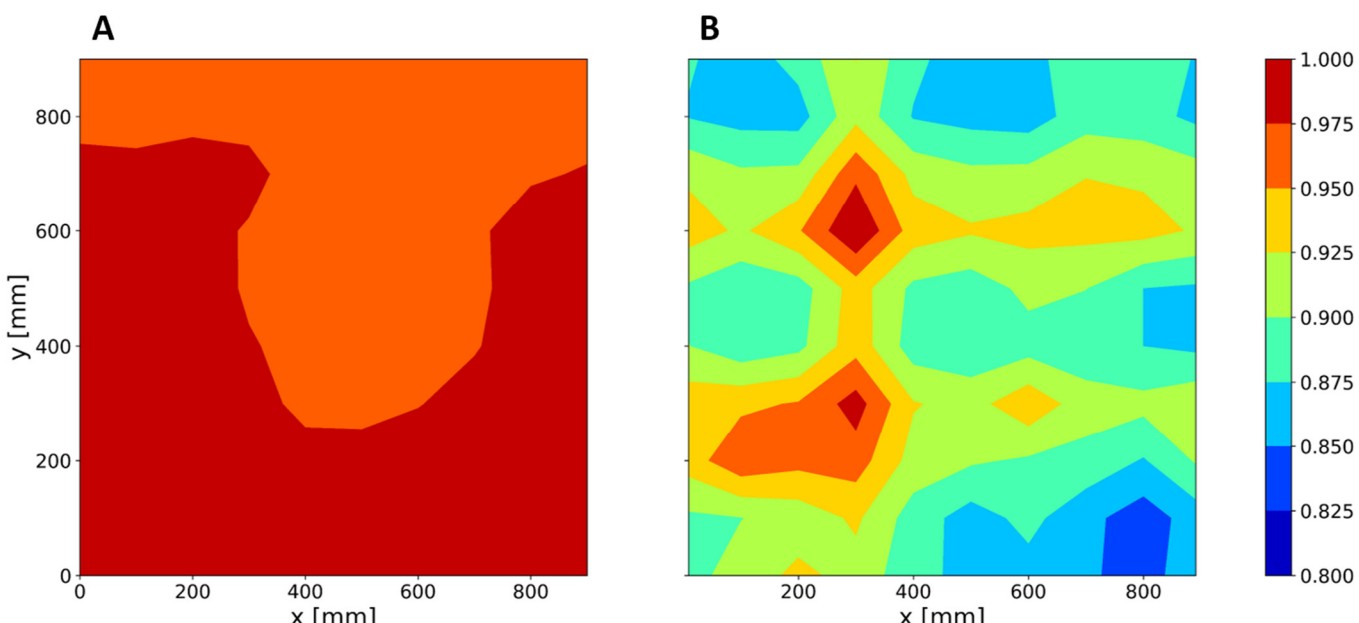

**Figure 5.** Relative R/B photon irradiance ratios for illuminance settings (**A**),(**B**). The R/B ratio is more uniform in (**A**) than in (**B**) as a result of more effective color mixing in case of setting (**A**) compared to (**B**).

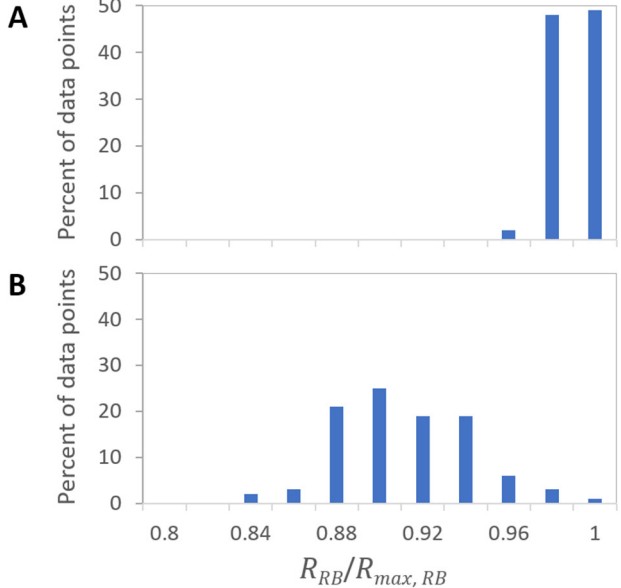

**Figure 6.** Relative distribution of Red/Blue photon irradiance ratios for illuminance settings (**A**),(**B**). *RR/B* values were normalized to the maximum value, *R*max. The width of the PPFD distribution is narrower in (**A**) compared to (**B**) in line with the contour plots in Figure 5.

## 4. Discussion

We described measurements and procedures for data analysis to benchmark and compare the lighting installations. There are, however, several limitations to using the photon irradiance distribution parameters directly as a figure of merit in the performance assessment of horticultural lighting applications. Intuitively, we know that the narrower the photon irradiance distribution is the better, but setting specification limits on $U_o$ or $U_d$, without considering the light response curve of the crop would be a source of confusion and decision error. In the following, we argue that $U_o$ being greater in case of (A) relative

to (B) cannot drive the conclusion automatically that (A) is better than (B). Depending on the actual crop cultivated, the performance of the lighting environment (B) can be close to (A) or much worse than the performance of (A).

In the following, we will rely on the assumption that all non-lighting environmental conditions are held constant in settings (A),(B). We also presume that the PPFD level as well as the photoperiod completely determine the rate of crop growth. An objective analysis also excludes any preconcept about the spectral features, such as the advantage or disadvantage of having elevated photon irradiance in the G band like in setting (B) or the multitude of narrow band emissions in setting (A). All photons in the PAR range received by the target area are considered to have equal impact.

Spatial variation of photon irradiance distribution over the cultivated crop can result in differences in the local plant growth rates. In addition to the local PPFD value, we need to take into account the quantitative light response of the crop grown under the tested illumination as well. Generally, the rate of photosynthesis increases with the increase of photon irradiance up to the light saturation point where the photosynthetic rate levels off. In the light saturation regime, the photosynthetic activity is limited by factors other than the light intensity, such as the $CO_2$ concentration. Assuming all other growth factors are held constant, including the spectral distribution of photon irradiance, the crop yield is proportional to the rate of photosynthesis and shows a photo-response similar to the photosynthetic light response curves [14].

The economics of indoor cultivation with electric light is a balance between the energy cost and the generated crop value [38,39]. The energy cost increases linearly with the average PPFD, but the crop value is close to a step function of the irradiance. Depending on the actual crop growth transfer function, a specific illumination setting and the related overall performance of the lighting system may be acceptable or unsatisfactory for the grower.

We can directly connect the crop light response curve with the relative PPFD distributions in Figure 4 to predict the yield of the cultivation system. Plants grown under different lighting conditions will represent a different commercial value by the end of the cultivation cycle. In our analysis, we consider two cases: the crop is sold on some quantitative parameter, such as weight, or amount of aromatic content. In this case, there is a continuous monotonic function connecting the commercial value with the relative photon irradiance. In the other case, we consider products going through a go/no go quality testing. If the product meets the acceptance criteria, e.g., height, number of leaves, color, etc., the product will be sold at the same price independently from the individual differences. The product will be discarded if it fails the quality inspection, and its commercial value will be considered to be zero.

The entitlement of the commercial value generated in a cultivation cycle corresponds to the crop value produced if all the points of the work plane were illuminated by the target photon irradiance. Any deviation from the perfectly uniform illumination will lower the economics of the cultivation system. We limit our discussion to the case where the maximum of the PPFD distribution is set to the photosynthetic photon irradiance at the light saturation point. The rationale in this case is that plants under the maximum photon irradiance will develop at the highest rate. All other plants will develop slower depending on the local photon irradiance level at their position.

In Figure 7, the relative commercial value, $f(E_r)$, is plotted as a function of the relative photon irradiance $(E_r = E/E_t)$, where the photon irradiance $E$ is normalized to the target photon irradiance, corresponding to the light saturation point $E_t$. The dotted blue line represents the continuous commercial value transfer function $f_1$, whereas the dotted gray line $f_2$ is an example of a quality inspection transfer function assuming that a crop grown below 80% of the target photon illuminance has no commercial value. The dotted blue line was a fit for real fresh weight vs. PPFD data from a lettuce growth experiment [40]. This function was discretized (solid blue line) for the same relative PPFD bins used in the assessment of photon irradiance distributions shown in Figure 4A,B.

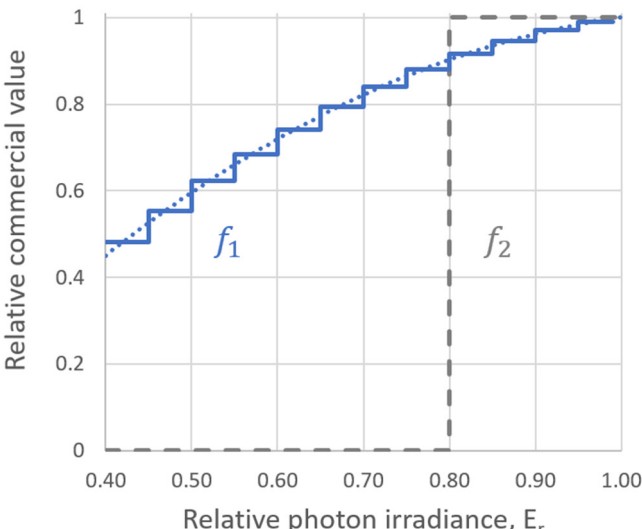

**Figure 7.** Relative commercial value of the crop as a function of photosynthetic photon irradiance normalized to the target value. Solid blue line ($f_1$) represents the case when commercial value changes continuously as a function of photon irradiance (e.g., fresh weight of the crop). The gray dashed line ($f_2$) is the transfer function when crop is sold by piece based on quality attributes.

The figure of merit from the horticultural application perspective is the economic value produced relative to the entitlement. We define this efficiency metric as the utilization of the illuminated area, $\eta_Y$, expressed by

$$\eta_Y = 100\% \times \sum_{E_r=0}^{1} f(E_r)h(E_r) \tag{3}$$

where $h(E_r)$ stands for the histogram of the relative photosynthetic photon irradiance distribution, and $f(E_r)$ is the transfer function between the relative crop yield and the relative photosynthetic photon irradiance. $\eta_Y$ can be regarded as a weighted average of the crop yield where the weighting factor is the PPFD distribution. The utilization of the illuminated area has the highest value, 100%, if photon irradiance is at the target value at all points of the illuminated surface. We want to highlight that this efficiency metric characterizes the photon irradiance distribution only; it does not take into account the total photon flux emitted by the luminaires.

In Table 3, $\eta_Y$ values calculated for the transfer functions $f_1$ and $f_2$ highlight the transfer function dependence of the utilization factor. Despite the striking difference in the photon irradiance distributions, the relative crop yield between illumination settings (**A**),(**B**) is moderate, 96% vs. 87%. On the other hand, there is a factor 2 difference between the two illumination settings (99% vs. 46%) if there is a hard product acceptance threshold at 0.8 relative photon irradiance.

**Table 3.** Utilization of illuminated area ($\eta_Y$) calculated for illumination settings (**A**),(**B**) and for transfer functions $f_1$ and $f_2$.

| Illumination Setting | (A) | | (B) | |
|---|---|---|---|---|
| Transfer function | $f_1$ | $f_2$ | $f_1$ | $f_2$ |
| Utilization of illuminated area, $\eta_Y$ | 96% | 99% | 87% | 46% |

These results demonstrate that the lighting parameters alone do not provide sufficient information to benchmark and compare the performance of indoor cultivation systems using LED light. To optimize productivity, the light response of the crop and several

economic parameters, i.e., electricity price and market value of the crop, should also be incorporated into the model of controlled environment production systems.

In our discussion, we calculated with a one variable transfer function having PPFD as an input. For a more complete analysis, taking into account the spectral variations of the illumination, a multivariate function of B, G, R and FR photon irradiance values would be required. Experimental determination of these multivariate transfer functions is an upcoming challenge for horticultural research.

## 5. Conclusions

Controlling lighting parameters in indoor farming is of ultimate importance to maximize crop production efficiency. In our study, we highlighted the advantage of carrying out high spatial resolution photon irradiance measurements to characterize the lighting environment. We relied on several simplifications: measurements were restricted to one horizontal plane only and did not address vertical penetration of light into the plant canopy. It was assumed that light arriving on plant leaves is fully utilized, and in the light response of plants, we did not address the impact of many other parameters affecting plant growth. To clarify the interaction of all other environmental parameters with the lighting conditions, further targeted studies are necessary.

**Author Contributions:** Conceptualization, L.B. and L.S.; methodology, L.B. and L.S.; validation, L.B., L.S. and L.C.; investigation, L.B. and Z.D.; resources, L.B.; writing—original draft preparation, L.B., L.S. and L.C.; writing—review and editing, L.B., L.S., L.C. and Z.D.; visualization, L.B.; supervision, L.B. and L.S. All authors have read and agreed to the published version of the manuscript.

**Funding:** This research received no external funding.

**Institutional Review Board Statement:** Not applicable.

**Informed Consent Statement:** Not applicable.

**Data Availability Statement:** Not applicable.

**Acknowledgments:** Not applicable.

**Conflicts of Interest:** The authors declare no conflict of interest.

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
