# Peer review of "Characterizing the Spatial Uniformity of Light Intensity and Spectrum for Indoor Crop Production"

_horticulturae, doi:10.3390/horticulturae8070644_

Round 1
Reviewer 1 Report
The conclusion is too long and it should be concise and predominantly mentioned the core findings of the current study.
Author Response
Dear Reviewer1,
Thank you for your constructive positive comments which help us to improve the quality of our paper. Please find below our answers and actions regarding your questions and concerns:
The keywords must be changed and add new lucrative keywords that are not related to the title of the research.
We updated the keywords: sole-source lighting, spectroradiometer, lighting characteristics, crop growth model, vertical farm productivity
The introduction need to improved and incorporated recent relevant information regarding the experiment and mainly focused on research gap.
The “Introduction” has been upgraded. The following new references related to the spatial distribution of light were included.
- Xu, Y.; Wang, H.; Nsengiyumva, W. Analysis of the Uniformity of Light in a Plant Growth Chamber. In 2018 4th International Conference on Universal Village (UV); 2018; pp 1–7. https://doi.org/10.1109/UV.2018.8642131.
- SAITO, Y.; SHIMIZU, H.; NAKASHIMA, H.; MIYASAKA, J.; OHDOI, K. Effect of Distribution of Photosynthetic Photon Flux Density Created by LEDs and Condenser Lenses on Growth of Leaf Lettuce (Lactuca Sativa Var. Angustana). Environmental Control in Biology 2013, 51 (3), 131–137. https://doi.org/10.2525/ecb.51.131.
- Barceló-Muñoz, A.; Barceló-Muñoz, M.; Gago-Calderon, A. Effect of LED Lighting on Physical Environment and Microenvironment on In Vitro Plant Growth and Morphogenesis: The Need to Standardize Lighting Conditions and Their Description. Plants 2022, 11 (1). https://doi.org/10.3390/plants11010060.
- Paucek, I.; Appolloni, E.; Pennisi, G.; Quaini, S.; Gianquinto, G.; Orsini, F. LED Lighting Systems for Horticulture: Business Growth and Global Distribution. Sustainability 2020, 12 (18). https://doi.org/10.3390/su12187516.
- Batista, D. S.; Felipe, S. H. S.; Silva, T. D.; de Castro, K. M.; Mamedes-Rodrigues, T. C.; Miranda, N. A.; Ríos-Ríos, A. M.; Faria, D. V.; Fortini, E. A.; Chagas, K.; Torres-Silva, G.; Xavier, A.; Arencibia, A. D.; Otoni, W. C. Light Quality in Plant Tissue Culture: Does It Matter? In Vitro Cellular & Developmental Biology - Plant 2018, 54 (3), 195–215. https://doi.org/10.1007/s11627-018-9902-5.
- Llewellyn, D.; Shelford, T.; Zheng, Y.; Both, A. J. Measuring and Reporting Lighting Characteristics Important for Controlled Environment Plant Production. In Acta Horticulturae; International Society for Horticultural Science (ISHS), Leuven, Belgium, 2022; pp 249–254. https://doi.org/10.17660/ActaHortic.2022.1337.34.
- Ke, X.; Yoshida, H.; Hikosaka, S.; Goto, E. Optimization of Photosynthetic Photon Flux Density and Light Quality for Increasing Radiation-Use Efficiency in Dwarf Tomato under LED Light at the Vegetative Growth Stage. Plants 2021, 11 (1), 121. https://doi.org/10.3390/plants11010121.
- Pan, J., Li, Q., Li, X., Wen, Y. Spatial light distribution characterization and measurement of LED horticultural lights. In Proceedings of 29th CIE Session (Washington, USA: CIE), 2019, p.333–341. https://doi.org/10. 25039/x46.2019.OP46.
The need for a standardized lighting characterization procedure is now emphasized in the text.
The conclusion is too long and it should be concise and predominantly mentioned the core findings of the current study
We reworked and trimmed the “Conclusion”.

Reviewer 2 Report
The article Characterizing the spatial uniformity of photon irradiance distribution in controlled-environment agriculture is well structured, although there are some aspects that need to be improved and some corrections and/or observations that the authors should consider related to improving the article.
I find the article very interesting. The introduction and conclusions realistically set out the problems the world is facing right now and propose solutions.
Authors should clarify these aspects.
L71-72. I agree with the recommendation of the International Commission on illumination.
As the authors of this article state, it is necessary to know the irradiance per nanometer, but the authors use formulas whose results are expressed in micromol s -1 m -2 and give results in %.
L 121-122. It would be interesting to indicate the watts of each led peak in order to understand the spectral distribution of figure 2.
L. 127. They do not show the emission w of the leds of option A: 410, 435, 445, 465, 495, 595, 605, 630, 122, 660, 675, 735 nm.
Remarks on Figure 2.
Plots are of relative spectral intensity. My opinion is that from a photobiological point of view they do not represent any utility. They must be of irradiance, that is, the W/m2/nm that actually reach the plants.
In the results of figure 2, is the irradiance of the white LED included? If so, how significant is the white LED in the relative spectral distribution in A and B? How is it possible that the yellow emission is not representative?
Is the main weight on the 450nm peak because of the white led?
Is the 650nm peak so high in both cases because that led is higher power?
The curves are not apparently very different and there is no significant difference in light peaks between A and B. The authors consider that the performance is very different between lighting A and lighting B. Where would the difference be if the two main peaks are the same in A and B. Would the difference in irradiance of the two figures be between 500 and 600 nm? What is the real irradiance difference in that range?
Do not the authors of this article consider that in order to defend against system (B), their system (A) serves for a good optimization of the crop by having so many different led peaks in the curve, it would be necessary to be able to appreciate all the 410 led peaks, 435, 445, 465, 495, 595, 605, 630, 122 660, 675, 735?
L 163-164. I do not understand why the maximum peak has to be normalized to facilitate the comparison. It is enough to put on the x-axis the W corresponding to each nm of the y-axis. If this is not done, it is not possible to know how many watts (10-2, 10-3, 10-4, 10-5 W/m 2 /nm?) reach the crop and therefore discuss whether that amount of energy can be photobiologically significant. Can your spectroradiomer meausure real irradiance in the range of 500-600 nm at the distance of the crop?
L 382-385. This comment is logical but explain with the spectral distribution of the prototype shown in figure 2 how it could be obtained.
Their ideas are very positive and, in my opinion, very important to take into account them in the present, but I think that their study does not show what they want to defend. Changes are needed in the graphical representation of your results and explain them.
Author Response
Dear Reviewer2,
Thank you for your comments and proposals to improve the quality of our paper. Please find below our answers and list of actions (in black) inserted in your comments (blue):
The article Characterizing the spatial uniformity of photon irradiance distribution in controlled-environment agriculture is well structured, although there are some aspects that need to be improved and some corrections and/or observations that the authors should consider related to improving the article.
I find the article very interesting. The introduction and conclusions realistically set out the problems the world is facing right now and propose solutions.
Authors should clarify these aspects.
L71-72. I agree with the recommendation of the International Commission on illumination.
As the authors of this article state, it is necessary to know the irradiance per nanometer, but the authors use formulas whose results are expressed in micromol s -1 m -2 and give results in %.
Figure 2. was reworked to show irradiance spectra in absolute value: µmol s -1 m -2.
L 121-122. It would be interesting to indicate the watts of each led peak in order to understand the spectral distribution of figure 2.
- 127. They do not show the emission w of the leds of option A: 410, 435, 445, 465, 495, 595, 605, 630, 660, 675, 735 nm.
The luminaires we used are commercial products and the manufacturers did not provide LED chip level information. To answer the questions exactly reverse engineering of the products would be required. The updated Figure 2, however, provides accurate information on the spectral distribution of irradiance resulting from the superposition of individual LED emissions.
For a better understanding I have attached Figure A1 showing the 12 individual LED spectra of luminaire A. In case of luminaire A, it was possible to record individual spectra by placing the sensor head of the spectroradiometer close to the secondary optics of the LED and by blocking the light emission of other LEDs by a tubular shade element. The individual LED spectra were normalized to their maximum value.
Figure A 1. Individual LED spectra of luminaire A. The spectral distributions were normalized to the maximum spectral irradiance, respectively. Legend shows the nominal wavelength of the peak maxima. The emission spectrum of shown in Figure 2 are the superposition of these individual spectral distributions.
Remarks on Figure 2.
Plots are of relative spectral intensity. My opinion is that from a photobiological point of view they do not represent any utility. They must be of irradiance, that is, the W/m2/nm that actually reach the plants.
Figure 2. was reworked to show irradiance spectra in absolute value: µmol s -1 m -2.
In the results of figure 2, is the irradiance of the white LED included? If so, how significant is the white LED in the relative spectral distribution in A and B? How is it possible that the yellow emission is not representative?
In luminaire B the proportion of white LEDs is higher compared to luminaire A. Irradiance within the 500-600 nm band is coming from the white LEDs in case B, whereas in case A, the major sources of photon irradiance in band G are the 495 nm, 595 nm and 605 nm narrow band LEDs .Table 1 shows quantitative information of photon irradiance in specific wavebands.
Is the main weight on the 450nm peak because of the white led?
White LEDs comprise blue light emitting diodes covered by a phosphor layer, therefore white LEDs have a strong emission at 450 nm. On top of that additional blue LEDs are also on the LED panel. 450 nm irradiance is a superposition of the irradiances coming from the blue 450 nm and white LEDs.
Is the 650nm peak so high in both cases because that led is higher power?
The reason is the higher number (and total power) of deep red LEDs in the luminaires compared to other LEDs with other wavelengths.
The curves are not apparently very different and there is no significant difference in light peaks between A and B. The authors consider that the performance is very different between lighting A and lighting B. Where would the difference be if the two main peaks are the same in A and B. Would the difference in irradiance of the two figures be between 500 and 600 nm? What is the real irradiance difference in that range?
The modified Figure 2 shows the spectral irradiance in absolute values. The irradiance in the 500-600 nm range is higher in case of luminaire B because this luminaire contains more white LEDs.
Do not the authors of this article consider that in order to defend against system (B), their system (A) serves for a good optimization of the crop by having so many different led peaks in the curve, it would be necessary to be able to appreciate all the 410 led peaks, 435, 445, 465, 495, 595, 605, 630, 660, 675, 735?
Please note that the wavelength values refer to the position of the maximum wavelength of the individual LED emissions. The emission spectra of individual LEDs overlap resulting in the spectra presented in Figure 2. The 11 different wavelengths do no show up as separate peaks. One can recognize shoulders at the wavelengths related to individual spectra in Figure 2A as shown below.
L 163-164. I do not understand why the maximum peak has to be normalized to facilitate the comparison. It is enough to put on the x-axis the W corresponding to each nm of the y-axis. If this is not done, it is not possible to know how many watts (10-2, 10-3, 10-4, 10-5 W/m 2 /nm?) reach the crop and therefore discuss whether that amount of energy can be photobiologically significant. Can your spectroradiomer meausure real irradiance in the range of 500-600 nm at the distance of the crop?
Figure 2. was reworked to show irradiance spectra in absolute value: µmol s -1 m -2. Table 1 also summarizes photon irradiance values in the B, G, R, FR and PAR wavebands.
L 382-385. This comment is logical but explain with the spectral distribution of the prototype shown in figure 2 how it could be obtained.
In the yield calculation we assumed that the crop growth is determined entirely by the PPFD values and contributions from spectral differences (including differences in photon irradiance in the G waveband) between A and B can be neglected. The PPFD values were determined by scaling the spectral irradiance distributions shown in Figure 2 and integrating between 400 and 700 nm according to Equation 1. To take into account spectral differences we would need a multivariate transfer function between the characteristics of spectral irradiance distribution (e.g. photon irradiances in B, G, R and FR) and crop yield. To measure such a transfer function increases complexity of the experimental work. The scope of our research is to find the tradeoff between the simplicity and accuracy of the model we use to predict the productivity of indoor cultivation systems.
Their ideas are very positive and, in my opinion, very important to take into account them in the present, but I think that their study does not show what they want to defend. Changes are needed in the graphical representation of your results and explain them
Figures 2-5 and related explanations were upgraded.

Reviewer 3 Report
General comments
The title doesn’t quite capture the main topic discussed in the manuscript. For example, controlled environment agriculture includes greenhouse production. Greenhouses can use a combination of sunlight and supplemental light to grow crops. However, the manuscript focuses on indoor crop production, using electric lighting (often termed sole-source lighting). The title also doesn’t make it clear that your analysis focused on specific wavebands (B, G, R, and FR).
The wavelength ranges you used to define B, G, R, and FR include wavelengths that belong to different waveband (500, 600, and 700 nm). For example, you define 500 nm as both B and G. It’s better to define the ranges as 400-499, 500-599, 600-699, 700-800 nm.
Your analysis is focused on short crops (e.g., leafy greens). However, light uniformity in tall crops (e.g., vine crops) is a much more challenging issues and this should be addressed in more detail.
Figure and table captions should end with a period.
A useful uniformity metric is defined as 1 – (st. dev./average). It has long been suggested (see several of the older references in your list) that the min/max or the min/average are not particularly useful because they put substantial weight on the extreme values in a set of measurements. I’m disappointed that you still promote these metrics because we’ve been trying to get the horticultural lighting industry to stop using them.
Illuminance setting A is not feasible for vertical farming operations due to the mounting height above the crop (1.6 m). The difference in mounting height between illuminance settings A and B and the different types of luminaires used for these different settings results in predictable differences in uniformity. Nevertheless, it is curious that the R/B ratios shown in Figure 5B are not more uniform across the measurement area.
There is a difference between the number of photons delivered to a crop and the number of photons used by that crop (for photosynthesis). This difference is not addressed in your manuscript.
Crop parameters (e.g., weight) are determined by more than just the light conditions. This could be better explained in your manuscript.
I found your discussion around Figure 7 difficult to follow. This was probably due to the fact that you use relative (normalized) values in that figure.
Please check the formatting of your references. For example, number 6 is missing an initial, and the title of number 25 is incorrect.
Specific comments
Controlled environment agriculture is typically written without the dash between controlled and environment (and can be abbreviated as CEA).
Please change the term artificial lighting in electric lighting. There’s nothing artificial about electric lighting.
Line 61: Some crops need to be protected (not all).
Line 75: Replace ‘deep-rooted in’ with ‘common in the’.
Line 83: replace ‘it’ with ‘this’.
Line 86: Note that a specific lamp type (e.g., HPS) produced by different manufacturers can have small differences in their spectral output.
Line 90: And the DLI?
Materials and Methods: Please provide more detail about fixture performance (e.g., wattage, PFD output, efficacies, optics). Did you use a warm-up period before collecting irradiance data? Please include the model numbers for all equipment/instruments used.
Line 128: Add a comma after 660.
Line 132: Approximately is not a scientifically repeatable term.
Line 163: The subscript seems too small (same for line 167).
Equation 1: Do you need the wavelength immediately after the summation symbol?
Figure 3: The font sizes for the axes is too small.
Line 262: Subscript for P10?
Figure 4: Replace the small letters a and b with capital letters (remove the parentheses).
Table 3: Create a better separation between the data for illuminance settings A and B.
Line 311: Change last part of the sentence to read: ‘…values could have a substantial effect on crop growth.’
Line 320: remove ‘the’ in front of application engineers.
Line 336: Change to: ‘…crop can result in…’
Line 371: Delete ‘gauge testing of’
Line 373: Replace ‘scrapped’ with ‘discarded’
Line 393: Insert ‘the’ in front of relative (two times).
Line 401: Is the figure number correct? Replace ‘if’ with ‘shown in’
Line 405: Equation number = 3
Line 414: Table number = 4 (same for line 421).
Line 431: Replace ‘model of CEA grow systems’ with ‘assessment of controlled environment production systems’
Line 439: Delete ‘in’
Line 440: Delete ‘the’
Line 445: replace ‘reduces’ with ‘can reduce’
Line 458: Start with ‘In a CEA production facility, microgreens…’
Line 462: Delete ‘to conduct’
Author Response
Dear Reviewer3,
Thank you for your detailed and insightful comments which help us to improve the quality of our paper. Please find below our answers and list of actions (black) regarding your questions and concerns (blue):
General comments
The title doesn’t quite capture the main topic discussed in the manuscript. For example, controlled environment agriculture includes greenhouse production. Greenhouses can use a combination of sunlight and supplemental light to grow crops. However, the manuscript focuses on indoor crop production, using electric lighting (often termed sole-source lighting). The title also doesn’t make it clear that your analysis focused on specific wavebands (B, G, R, and FR).
We changed the title to: Characterizing the spatial uniformity of light intensity and spectrum for indoor crop production
The wavelength ranges you used to define B, G, R, and FR include wavelengths that belong to different waveband (500, 600, and 700 nm). For example, you define 500 nm as both B and G. It’s better to define the ranges as 400-499, 500-599, 600-699, 700-800 nm.
We agree and confirm that the upper limit of the interval has not been included in the calculation. The waveband definitions were updated with comments that this definition assumes that spectra are interpolated to integer wavelength values with 1 nm resolution.
Your analysis is focused on short crops (e.g., leafy greens). However, light uniformity in tall crops (e.g., vine crops) is a much more challenging issues and this should be addressed in more detail.
We included address 3D challenges in the introduction and conclusion as a next step for our research. In this study we focused on horizontal distribution of light.
Figure and table captions should end with a period.
Period has been inserted in each caption.
A useful uniformity metric is defined as 1 – (st. dev./average). It has long been suggested (see several of the older references in your list) that the min/max or the min/average are not particularly useful because they put substantial weight on the extreme values in a set of measurements. I’m disappointed that you still promote these metrics because we’ve been trying to get the horticultural lighting industry to stop using them.
We have included the complement of the coefficient of variance (1-CV) in Tables 1 and 2.
Illuminance setting A is not feasible for vertical farming operations due to the mounting height above the crop (1.6 m). The difference in mounting height between illuminance settings A and B and the different types of luminaires used for these different settings results in predictable differences in uniformity. Nevertheless, it is curious that the R/B ratios shown in Figure 5B are not more uniform across the measurement area.
Luminaire A utilizes secondary optics collimating the light of individual LEDs. This may be the reason why the uniformity is worse relative to your expectation. The secondary optics increases the amount of useful photon flux arriving at the target area. We address the tradeoff between uniformity and utility in another publication. Our objective was to compare two lighting distributions and agree with your comment that the illumination setting A is not feasible for vertical farming. We included a sentence that Luminaire A was originally designed to replace 1000 W HPS applications.
There is a difference between the number of photons delivered to a crop and the number of photons used by that crop (for photosynthesis). This difference is not addressed in your manuscript.
Quantum yield of photosynthesis was beyond the scope of the paper but we included comment on this in the reworked Conclusion.
Crop parameters (e.g., weight) are determined by more than just the light conditions. This could be better explained in your manuscript.
Addressing non-lighting parameters was beyond the scope of this paper but we included comment on this in the reworked Conclusion.
I found your discussion around Figure 7 difficult to follow. This was probably due to the fact that you use relative (normalized) values in that figure.
We reworked the explanation.
Please check the formatting of your references. For example, number 6 is missing an initial, and the title of number 25 is incorrect.
References have been corrected and updated.
Specific comments
Controlled environment agriculture is typically written without the dash between controlled and environment (and can be abbreviated as CEA).
corrected
Please change the term artificial lighting in electric lighting. There’s nothing artificial about electric lighting.
done
Line 61: Some crops need to be protected (not all). done
Line 75: Replace ‘deep-rooted in’ with ‘common in the’. done
Line 83: replace ‘it’ with ‘this’. done
Line 86: Note that a specific lamp type (e.g., HPS) produced by different manufacturers can have small differences in their spectral output. We agree and assumed that the minor difference is negligible in this discussion.
Line 90: And the DLI? DLI included
Materials and Methods: Please provide more detail about fixture performance (e.g., wattage, PFD output, efficacies, optics). Did you use a warm-up period before collecting irradiance data? Please include the model numbers for all equipment/instruments used.
Additionai information provided.
30 min warm-up was used, included in the text,
Line 128: Add a comma after 660. done
Line 132: Approximately is not a scientifically repeatable term. corrected
Line 163: The subscript seems too small (same for line 167). corrected
Equation 1: Do you need the wavelength immediately after the summation symbol?
Figure 3: The font sizes for the axes is too small. corrected
Line 262: Subscript for P10? corrected
Figure 4: Replace the small letters a and b with capital letters (remove the parentheses). corrected
Table 3: Create a better separation between the data for illuminance settings A and B. corrected
Line 311: Change last part of the sentence to read: ‘…values could have a substantial effect on crop growth.’ corrected
Line 320: remove ‘the’ in front of application engineers. corrected
Line 336: Change to: ‘…crop can result in…’ done
Line 371: Delete ‘gauge testing of’ done
Line 373: Replace ‘scrapped’ with ‘discarded’ corrected
Line 393: Insert ‘the’ in front of relative (two times). done
Line 401: Is the figure number correct? Replace ‘if’ with ‘shown in’ done
Line 405: Equation number = 3 done
Line 414: Table number = 4 (same for line 421). Corrected, should be 3
Line 431: Replace ‘model of CEA grow systems’ with ‘assessment of controlled environment production systems’ done
Line 439: Delete ‘in’ done
Line 440: Delete ‘the’ done
Line 445: replace ‘reduces’ with ‘can reduce’ done
Line 458: Start with ‘In a CEA production facility, microgreens…’ done
Line 462: Delete ‘to conduct’ done

Round 2
Reviewer 2 Report
The article Characterizing the spatial uniformity of photon irradiance distribution in controlled-environment agriculture is well structured, although there are some aspects that need to be improved and some corrections and/or observations that the authors should consider related to improving the article.
Authors should clarify these aspects.
Remarks on Figure 2.
Plots are of relative spectral intensity. My opinion is that from a photobiological point of view they do not represent any utility. They must be of irradiance, that is, the W/m2/nm that actually reach the plants.
In the results of figure 2, is the irradiance of the white LED included? If so, how significant is the white LED in the relative spectral distribution in A and B? How is it possible that the yellow emission is not representative?
Is the main weight on the 450nm peak because of the white led?
Is the 650nm peak so high in both cases because that led is higher power?
The curves are not apparently very different and there is no significant difference in light peaks between A and B. The authors consider that the performance is very different between lighting A and lighting B. Where would the difference be if the two main peaks are the same in A and B. Would the difference in irradiance of the two figures be between 500 and 600 nm? What is the real irradiance difference in that range?
Do not the authors of this article consider that in order to defend against system (B), their system (A) serves for a good optimization of the crop by having so many different led peaks in the curve, it would be necessary to be able to appreciate all the 410 led peaks, 435, 445, 465, 495, 595, 605, 630, 122 660, 675, 735?
L 163-164. I do not understand why the maximum peak has to be normalized to facilitate the comparison. It is enough to put on the x-axis the W corresponding to each nm of the y-axis. If this is not done, it is not possible to know how many watts (10-2, 10-3, 10-4, 10-5 W/m 2 /nm?) reach the crop and therefore discuss whether that amount of energy can be photobiologically significant. Can your spectroradiomer meausure real irradiance in the range of 500-600 nm at the distance of the crop?
It is necessary to answer these questions. I think it is necessary the irradiation of crops with policromatic led luminaires but their profits are not technically justified with your graphics and explanations.
Author Response
Dear Reviewer 2,
Thank you for taking your time and reviewing the reworked manuscript. From the fact that you have repeated many of your questions regarding Fig. 2 we understand that you do not regard the changes we made in the first round satisfactory. In the second version we devoted a new sub-topic (3.1 Analysis of spectral irradiance distributions) to explain the points you raised, reworked Figure 2, and significantly modified the “Introduction” and the “Discussion”. In the following we repeat your questions from the Review_2_Round_2 in a numbered list printed in blue, followed by our explanation and list of modifications we made in the manuscript printed in black. We sincerely hope you find these improvements satisfactory and the changes will increase the quality of the paper.
The article Characterizing the spatial uniformity of photon irradiance distribution in controlled-environment agriculture is well structured, although there are some aspects that need to be improved and some corrections and/or observations that the authors should consider related to improving the article.
Authors should clarify these aspects.
Remarks on Figure 2.
- Plots are of relative spectral intensity. My opinion is that from a photobiological point of view they do not represent any utility. They must be of irradiance, that is, the W/m2/nm that actually reach the plants.
We agreed and accepted your comment. In the manuscript reworked in round 1 we already had the correct axis title. “Spectral irradiance [W m-2 nm-1]. Unfortunately, there was a typo in our Answer1. Spectra in Fig. 2 show absolute values of the spectral irradiance distribution in unit W m-2 nm-1. You can recognize new changes in Fig. 2 based on your comments below.
- Is the main weight on the 450nm peak because of the white led?
- Is the 650nm peak so high in both cases because that led is higher power?
We incorporated the following paragraph in the sub-topic 3.1:
3.1 Analysis of spectral irradiance distributions
In Figure 2 arrows indicate the peak wavelengths of the discrete NB LED types incorporated into the luminaires. In setting A) there were 11 different types of NB LEDs and one type of white LED in the lighting fixture. The emission spectra of the individual NB LEDs have ~20 nm spread about the peak wavelength at full width of half maximum. The relative height of an individual NB LED emission reflects the proportion of the radiated power of the specific LED type relative to the total radiated power of the luminaire. The adjacent NB LED peaks overlap, resulting in a wide emission band exhibiting shoulders in the 400-500 nm and the 580-700 nm range.
The phosphor converted white LED has a narrow emission at 445 nm and a broad emission with a maximum at 567 nm. The 445 nm irradiance is a superposition of the irradiances coming from the 445 nm NB LED and the 445 nm emission of the phosphor converted white LED in Figure 2B). On top of the superposition of white and NB 445 nm irradiance the contributions from the NB LEDs with adjacent peak wavelengths should also be considered in Figure 2A).
The broad emission in the green waveband between 500-600 nm is exclusively due to the phosphor emission of white LEDs in setting B). In luminaire A) the tails of the 495 nm, 595 nm and 605 nm NB LED peaks have contribution to the G region along with some white phosphor emission. The high intensity in the 500-600 nm wavebands of luminaire B) relative to luminaire A) is due to the high proportion of white phosphor LEDs in the lightbars of setting B).
In Figures 2A) and B) the highest peak was located at 660 nm as a result of the high proportion of 660 nm NB LEDs in both lighting equipment. Far red emission at about 735 nm is apparent in both spectra with higher intensity in case of lighting B) relative to the luminaire A). Far red is outside the PAR range consequently photon irradiance beyond 700 nm is excluded from the PPFD calculation. Far red radiation affects the growth and development of many crops [21] therefore a recent publication proposes the extension of the PAR range by 50 nm to the 400-750 nm waveband [22].
Another approach towards a more detailed characterization of the lighting environment is to measure photon irradiance values in 100 nm wide wavebands between 400 nm and 800 nm as described above. In this way we use four independent quantitative parameters instead of a single PPFD figure to characterize the lighting conditions. The ratios of the photon irradiance values related to B, G, R and FR range carry information on the spectral distribution. Using four wavebands instead of the single PAR range increases the granularity of data and can be regarded as a reasonable trade-off between simplicity and accuracy of processing spectral variations.
- The curves are not apparently very different and there is no significant difference in light peaks between A and B. The authors consider that the performance is very different between lighting A and lighting B. Where would the difference be if the two main peaks are the same in A and B. Would the difference in irradiance of the two figures be between 500 and 600 nm? What is the real irradiance difference in that range?
First of all, we want to stress, that the difference we are focusing on is the point-to-point spatial variation of PPFD values on the target area. We put the two spectral distributions on the same plot as you requested. For the entire G band the average photon irradiance values are shown in Table 11.4 mmol× m−2×s-1 for luminaire A) and 30.5 mmol× m−2×s-1 for luminaire B) If you are looking at 550 nm there is factor 5.7 difference between the spectral irradiance of A) and B)
- Do not the authors of this article consider that in order to defend against system (B), their system (A) serves for a good optimization of the crop by having so many different led peaks in the curve, it would be necessary to be able to appreciate all the 410 led peaks, 435, 445, 465, 495, 595, 605, 630, 122 660, 675, 735?
We have reworked Figure 2 to highlight the individual NB wavelengths and provided detailed description of the spectra in
3.1 Analysis of spectral irradiance distributions
In Figure 2 arrows indicate the peak wavelengths of the discrete NB LED types incorporated into the luminaires. In setting A) there were 11 different types of NB LEDs and one type of white LED in the lighting fixture. The emission spectra of the individual NB LEDs have ~20 nm spread about the peak wavelength at full width of half maximum. The relative height of an individual NB LED emission reflects the proportion of the radiated power of the specific LED type relative to the total radiated power of the luminaire. The adjacent NB LED peaks overlap, resulting in a wide emission band exhibiting shoulders in the 400-500 nm and the 580-700 nm range.
- L 163-164. I do not understand why the maximum peak has to be normalized to facilitate the comparison. It is enough to put on the x-axis the W corresponding to each nm of the y-axis. If this is not done, it is not possible to know how many watts (10-2, 10-3, 10-4, 10-5 W/m 2 /nm?) reach the crop and therefore discuss whether that amount of energy can be photobiologically significant. Can your spectroradiomer meausure real irradiance in the range of 500-600 nm at the distance of the crop?
With this question you are referring to the original manuscript reworked after round 1. Figure 2 has been reworked showing absolute spectral irradiance in W m-2 nm-1.
It is necessary to answer these questions. I think it is necessary the irradiation of crops with policromatic led luminaires but their profits are not technically justified with your graphics and explanations.
It was not out objective to argue for or against the full spectrum or narrow band spectrum luminaires.
Lines 447-450:
“ An objective analysis also excludes any preconcept about the spectral features, like the advantage or disadvantage of having elevated photon irradiance in the G band like in setting B) or the multitude of narrow band emissions in setting A).”
Following the major rework of the manuscript we hope we were able to clarify all the points you raised and thank you again for your revealing questions.

Round 3
Reviewer 2 Report
The article Characterizing the spatial uniformity of photon irradiance distribution in controlled-environment can be published.